# NLRC4 GOF Mutations, a Challenging Diagnosis from Neonatal Age to Adulthood

**DOI:** 10.3390/jcm10194369

**Published:** 2021-09-24

**Authors:** Juliette Bardet, Noémie Laverdure, Mathieu Fusaro, Capucine Picard, Lorna Garnier, Sébastien Viel, Sophie Collardeau-Frachon, Jean-Marie De Guillebon, Isabelle Durieu, Clémence Casari-Thery, Guillaume Mortamet, Audrey Laurent, Alexandre Belot

**Affiliations:** 1Pediatric Nephrology, Rheumatology, Dermatology Unit, Hôpital Femme Mère Enfant, Hospices Civils de Lyon, 69677 Bron, France; de-guillebon-de-resnes.jm@chu-nice.fr (J.-M.D.G.); audrey.laurent@chu-lyon.fr (A.L.); 2Pediatric Hepatology, Gastroenterology and Nutrition Unit, Hôpital Femme Mère Enfant, Hospices Civils de Lyon, 69677 Bron, France; noemie.laverdure@chu-lyon.fr; 3Study Center for Primary Immunodeficiencies, AP-HP, Necker Hospital for Sick Children, Université Paris, 75015 Paris, France; mathieu.fusaro@aphp.fr (M.F.); capucine.picard@aphp.fr (C.P.); 4Immunology Department, Lyon Sud University Hospital, 69495 Pierre-Bénite, France; lorna.garnier@chu-lyon.fr (L.G.); sebastien.viel@chu-lyon.fr (S.V.); 5International Center of Research in Infectiology, Lyon University, INSERM U1111, CNRS UMR 5308, ENS, UCBL, 69007 Lyon, France; 6National Referee Centre for Rheumatic and AutoImmune and Systemic Diseases in Children (RAISE), 69677 Bron, France; clemence.casari-thery@chu-lyon.fr; 7Lyon Immunopathology Federation LIFE, Hospices Civils de Lyon, 69002 Lyon, France; 8Department of Pathology, Hospices Civils de Lyon-Hôpital Femme-Mère-Enfant, Claude Bernard Lyon 1 University, 69677 Bron, France; sophie.collardeau-frachon@chu-lyon.fr; 9Adult Cystic Fibrosis Center, Internal Medicine and Vascular Pathology Department, Groupement Hospitalier Lyon-Sud, Hospices Civils de Lyon, 69310 Pierre-Bénite, France; isabelle.durieu@chu-lyon.fr; 10Department of Internal and Vascular Medicine, Hôpital Lyon Sud, Hospices Civils de Lyon, 69310 Pierre-Bénite, France; 11Faculty of Medicine, University of Lyon, 69100 Villeurbanne, France; 12Pediatric Intensive Care Unit, Grenoble Alpes University Hospital, 38700 La Tronche, France; gmortamet@chu-grenoble.fr

**Keywords:** NLRC4, inflammasome, inflammasomopathy, sepsis, autoinflammatory diseases, allergy

## Abstract

The NLRC4 inflammasome is part of the human immune innate system. Its activation leads to the cleavage of pro-inflammatory cytokines IL-1β and IL-18, promoting inflammation. *NLRC4* gain-of-function (GOF) mutations have been associated with early-onset recurrent fever, recurrent macrophagic activation syndrome and enterocolitis. Herein, we describe two new patients with *NLRC4* mutations. The first case presented with recurrent fever and vasoplegic syndrome, gut symptoms and urticarial rashes initially misdiagnosed as a severe protein-induced enterocolitis syndrome. The second case had recurrent macrophage activation syndrome (MAS) and shock, suggesting severe infection. We identified two *NLRC4* mutations, on exon 4, within the nucleotide-binding protein domain (NBD). After a systematic review of *NLRC4* GOF mutations, we highlight the wide spectrum of this disease with a limited genotype–phenotype correlation. Vasoplegic shock was only reported in patients with mutation in the NBD. Diagnosing this new entity combined with gastrointestinal symptoms and vasoplegic shocks is challenging. It mimics severe allergic reaction or sepsis. The plasma IL-18 level and genetic screening are instrumental to make a final diagnosis.

## 1. Introduction

NLR-family CARD domain-containing protein 4 (NLRC4) acts as an innate sensor of the immune system. It is mainly expressed in macrophages and intestinal epithelial cells. Its capacity to promote the cleavage, maturation and secretion of interleukin-1β includes NLRC4 in the family of inflammasomes [1].

The *NLRC4* gene is located on chromosome 2p21-p22 and is made of nine exons. Its transcription is activated by the tumor suppressor protein p53 or pro-inflammatory stimuli such as TNFα [2]. It encodes a protein of 1024 amino acids with a molecular weight of 11 kDa. The NLRC4 protein has a N-terminal caspase activating and recruitment domain (CARD), a central nucleotide binding domain (NBD) and a leucine-rich repeat (LRR) domain. The CARD domain interacts with itself and other CARD proteins to oligomerize and the LRR domain is believed to be a regulatory domain since its removal results in a constitutionally active NLR protein [3].

The NLRC4 inflammasome responds to cytosolic flagellin, and to the inner rod and needle proteins of the type 3 secretion system of bacteria. Its oligomerization results in an exponential cascade of adapter recruitment and caspase-1 autoproteolysis known as inflammasome activation, resulting in the cleavage of the inflammatory substrates pro-IL-1β, pro-IL18, and gasdermin D. IL-1β mediates the classic signs and symptoms of the febrile response. IL-18 amplifies lymphocyte functions, increasing cytotoxicity through IFNγ production [2].

Aberrant activation of the NLRC4 inflammasome was recently identified in *de novo* gain-of-function (GOF) mutations in *NLRC4* in the context of macrophage activation syndrome (MAS), neonatal enterocolitis, fetal thrombotic vasculopathy, familial cold autoinflammatory syndrome and even death [4] but reports are scarce with only 12 cases identified in the literature.

Here, we report on two new cases of NLRC4 GOF expanding the spectrum of the disease and we discuss the clinical and immunological anomalies that are critical for the diagnosis.

## 2. Clinical Case Reports

### 2.1. Patient 1: An Early Digestive and systemic Inflammation with Shocks

Patient 1 was a girl with no past familial medical history, who presented from 20 days of life with recurrent episodes of fever with vasoplegic shocks associated with gastrointestinal symptoms (diarrhea, pain and bloating) and urticarial rash (Figure 1a).

Laboratory exams revealed features of MAS during the shocks (hyperferritinemia, trend to low fibrinogenemia, hypertriglyceridemia, pancytopenia) but the bone marrow aspiration did not show hemophagocytosis and the monocyte activation marker HLA-DR was at the lower limit, in contrast to the values usually seen in MAS. The assessment of interferon-stimulated genes in whole blood (the so-called interferon signature) was negative, while plasma IL-18 levels were dramatically elevated with values over 18,000 pg/mL (Figure 2). Abdominal ultrasound was normal.

Several diagnoses were evocated during the management of this young girl, including severe infection, inborn errors of metabolism and capillary leak syndrome.

Among autoinflammatory diseases, systemic-onset juvenile idiopathic arthritis and partial deficiency of mevalonate kinase were also evocated. Fecal calprotectin was normal, not in favor of early-onset inflammatory bowel disease. Severe allergic manifestations were also discussed and included cow’s milk protein allergy (CMPA), but the allergic tests were negative (RAST, specific IgE, histamine and serum tryptase at the time of the shocks). The second hypothesis was food protein-induced enterocolitis syndrome (FPIES), but methemoglobinemia was always negative and the hyperferritinemia made this diagnosis unlikely. In addition, one of the vasoplegic shocks occured without any food intake.

Gastrointestinal biopsies showed focal lesions of inflammation, subtotal to total villous atrophy of the duodenum without increased intraepithelial lymphocytes and mild chronic colitis. Intraepithelial eosinophilic infiltrates were also noticed in the duodenum and colon (Figure 1b–d).

Skin biopsy of a rash (Figure 1a) revealed neutrophil infiltrates and non-specific vasculitis lesions, without fibrinoid necrosis.

The hypothesis of a *NLRC4* GOF mutation was evoked at 3 months of life in light of the sustained inflammation with the elevated plasmatic IL-18 cytokine and the focal enteritis. Gene sequencing confirmed the diagnosis at 6 months of life with a new variant in the *NLRC4* gene on exon 4, variant c.620G > A (p.Arg207Lys). *In silico* prediction scores were in favor of a tolerated variant. This variant was not reported in public database dbSNP, gnomAD or HGMD.

Management was challenging and the child received several lines of antibiotics and intravenous immunoglobulin infusion with the hypothesis of an alloimmunization. Vasoplegic syndromes were treated by albumin infusion. Short cures of corticosteroids were started at 1.5 months of life with numerous retreatments during relapses. IL1 receptor antagonist (anakinra) was started at the age of 6 months as soon as the *NLRC4* mutation was identified. It generated a decrease in the number of attacks and severity, allowing steroid withdrawal and progressive food diversification. A switch was made to canakinumab two months later. After 15 months of treatment, there were no new episodes of flare (Figure 2).

### 2.2. Patient 2: Recurrent Macrophagic Activation Syndrome

Our second case was a teenager with recurrent MAS and shocks since the age of 3. At the time of diagnosis, she was 17 years old and had been hospitalized for 2 months for an acute febrile illness associated with non-malignant lymphoproliferation with multiple adenopathies and splenomegaly. Systemic-onset juvenile idiopathic arthritis was evocated.

Biologically, MAS was found with pancytopenia, cytolysis, hyperferritinemia up to 38,700 µg/L and mild hypofibrinogenemia at 1.61 g/L. IL-18 was not measured.

Infections were scrutinized but blood cultures remained sterile and viral and bacterial serologies returned negative except for EBV, in favor of a primary infection. Corticoids were initiated at the dose of 1 mg/kg/d and then 2 mg/kg/d (with the persistence of MAS), allowing a disappearance of lymph node enlargement and splenomegaly, and reduced frequency of febrile peaks. However, fever and cytopenia recurred as soon as the steroids were tapered, leading to two courses of etoposide therapy and then cyclosporine. After one month of treatment with corticosteroids and cyclosporine, with a positive EBV viral load, a treatment with four infusions of rituximab was performed, enabling the reduction of the viral load one month later.

She was then readmitted to the hospital as she presented shock associated with diffuse erythema and MAS despite treatment with corticosteroids and cyclosporine. Her condition required treatment with amines in intensive care. The infectious trigger found was *E. coli* pyelonephritis. Apyrexia was obtained after adapted antibiotic treatment. She then presented a new MAS episode associated with diarrhea, suggesting a probable gastrointestinal infectious trigger that resolved after increasing corticosteroid treatment.

Genetic sequencing reported a heterozygous variation classified as pathogenic of the NLRC4 gene: c.1010C > A, giving *p*.Thr337Asn, evocative for the NLRC4 GOF diagnosis. Mutation was already reported by Bracaglia et al. [5].

Her treatment with cyclosporine has since been gradually stopped, as well as the corticosteroids. The introduction of anti-IL1 or anti-IL18BP treatment will be considered if she develops MAS features or shock.

## 3. Literature Review

We conducted a literature review based on the PubMed database with the following keywords: inflammasome—NLRC4—gain of function—NLRC4 and autoinflammatory diseases up to 1 March 2021.

We identified 103 articles and 11 of those were reporting a total of 12 cases of NLRC4 GOF mutations (Figure 3 and Table 1).

Three previous studies [6,7,8] describe a de novo gain-of-function mutation of the *NLRC4* gene (c.1022 T > C, *p*.Val341Ala), with an alternative substitution for Barsalou et al. (c.1021G > C, *p*.Val341Leu). The phenotype encompassed neonatal-onset enterocolitis [6] or MAS [7,8].

Other mutations affecting the NBD were described, with variable phenotypes. MAS was described again with a de novo missense mutation (*p*. Thr337Ser or *p*. Thr337Asn) reported by Bracaglia et al. [5], Canna et al. [9] and in a 28-week preterm infant associated with hydrops with another mutation (p. Ser171Phe) [10].

Skin and rheumatic involvement were also observed by Kitamura and al. (c.1589A > C, *p*. His443Pro) [11], Volker-Touw et al. [12] and Wang et al. (c.514G > A (*p*.Gly172Ser)) [13]. Fever was not always present [12], although it appeared as a classical symptom.

Kawasaki et al. [14] studied the case of a Japanese male child diagnosed with NOMID without any *NLRP3* mutation. A WES revealed a somatic mosaicism of a novel *NLRC4* mutation c.529A > G, *p*. Thr177Ala.

Moghaddas et al. [15] reported the first situation of a mutation in the LRR domain of NLRC4. They showed that the mutation c. 1965 G > C, *p*. Thr655Cys NLRC4 increased inflammasome activation in vitro.

Chear et al. [16] reported a twelve-year-old Malay girl with recurrent fever, skin erythema and inflammatory arthritis. Whole exome sequencing (WES) and subsequent bidirectional Sanger sequencing identified a heterozygous missense mutation in NLRC4 (c.1970A > T). Cytokine analysis showed an extremely high serum IL-18 and IL-18/CXCL9 ratio, consistent with other NLRC4 GOF patients.

## 4. Discussion

Two distinct molecular mechanisms were reported in NLRC4 inflammasomopathy:Mutations within the NBD—Kitamura et al. [11] showed that a mutation in this domain promotes the oligomerization of the inflammasome with an increased response to usual stimuli.Mutations in LRR domain, described as a regulator domain. Its absence results in a constitutively active NLRC4 inflammasome [3].

Despite this molecular dichotomy, phenotypes are extremely varied. In fact, we find several MAS-like presentations in patients with mutations in the NBD or LRR domain (Case 1 and (6–10,14,15)). Milder phenotypes such as arthritis- or familial cold-induced autoinflammatory syndrome were described in NBD mutations [11,12,13,16] as for our patients with vasoplegic shocks. Therefore, the location of the genetic alteration is not predictive of the attack’s severity.

Our first case shows a well-known digestive manifestation in NLRC4 disease due to a high NLRC4 expression in epithelial intestinal cells. However, digestive presentation is also seen in several other situations including allergic manifestations. CMPA is often mentioned in infants’ rectal bleeding and removing cow’s protein from a child’s diet is the cornerstone of the treatment. Vasoplegic shocks are not described in CMPA. Conversely, food protein-induced enterocolitis syndrome (FPIES) is associated with shocks. Normal methemoglobinemia reduced the probability of this severe allergic manifestation [17] as well as the absence of a link to consuming a particular food or drink. Like systemic-onset JIA, NLRC4 GOF shares similarities with familial hemophagocytic lymphohistiocytosis. Indeed, the presence of splenomegaly, fever, gastrointestinal bleeding and rash at an early age could be consistent with this diagnosis. Furthermore, elevated IL-18 was also identified in HLH [18]. This latter fact complicates the diagnosis, especially since IL-18 is one of the promising diagnostic markers for *NLRC4* mutations [19]. Mevalonate kinase deficiency (MVK) is another autoinflammatory disease, known to also be associated with elevated IL-18 and gastrointestinal inflammation. IL-18 might also be elevated in systemic juvenile arthritis, thus the IL-18 level could help to differentiate autoinflammatory diseases or primary MAS (HLH) from other diseases, such as FPIES or other allergic or infectious severe diseases [20,21]. Patient 1 also mimicked idiopathic capillary leak syndrome. MAS is known to be a cause of secondary capillary leak syndrome, and the inflammatory phenomenon associated with cytokine storm points to secondary capillary leakage rather than idiopathic.

NLRC4 GOF shows a poor phenotype/genotype correlation and a wide spectrum of the symptoms. High IL-18 level may also contribute to the diagnosis and it has been described as correlating with MAS, particularly for free IL-18 [18] Genetic screening is the unique critical test enabling the final diagnosis.

## 5. Conclusions

Here, we report two cases of NLRC4 GOF associated with recurrent shocks. The wide spectrum of this disease from gastrointestinal symptoms to vasoplegic shocks demonstrates how the diagnosis can be challenging and illustrates that it can mimic severe allergic manifestations, sepsis or capillary leak syndrome. Diagnosis is challenging and plasma IL-18 level and subsequent genetic screening are instrumental to make the final diagnosis.

## Figures and Tables

**Figure 1 jcm-10-04369-f001:**
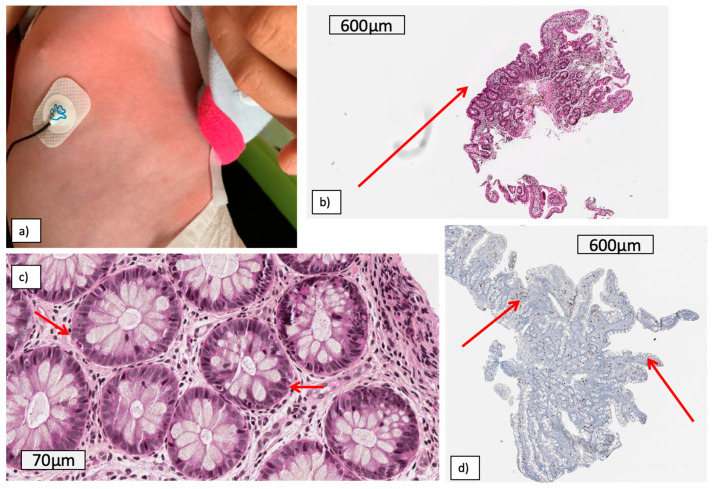
Skin rash (**a**) and gastrointestinal lesions (**b**–**d**) of patient 1: (**b**) duodenum: total to subtotal focal villous atrophy, (**c**) colon intraepithelial eosinophilic infiltrate, (**d**) duodenum: CD8 immunostaining: no increased intraepithelial lymphocytes (CD8).

**Figure 2 jcm-10-04369-f002:**
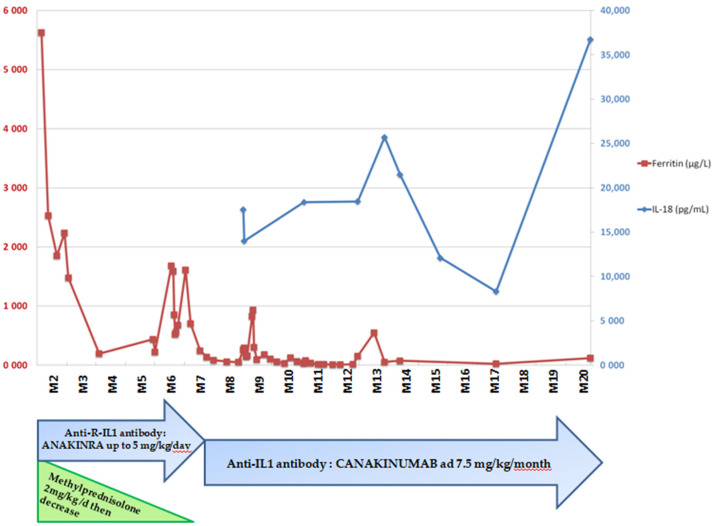
IL-18 and ferritin levels over time and management of patient 1.

**Figure 3 jcm-10-04369-f003:**
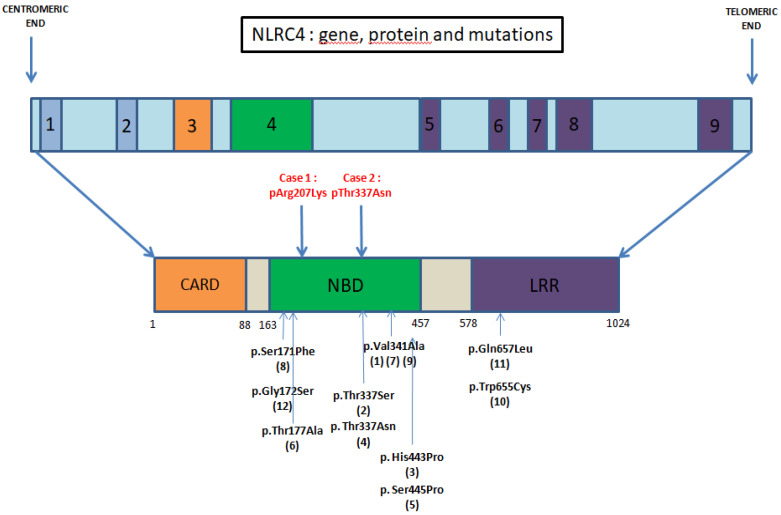
NLRC4 gene, protein and GOF mutations clinically described in the literature. [6] Romberg et al., 2014, [9] Canna et al., 2014, [11] Kitamura et al., 2014, [5] Bracaglia et al., 2015, [12] Volker-Touw et al., 2017, [14] Kawasaki et al., 2017, [7] Canna et al., 2017, [10] Liang et al., 2017, [8] Barsalou et al., 2018 (variant mutation pVal341Leu), [15] Mohggadas et al., 2018, [16] Chear et al., 2019, [13] Wang et al., 2020.

**Table 1 jcm-10-04369-t001:** Phenotype of patients according to *NLRC4* mutations.

*NLRC4* Gof Mutation	Symptoms							Lab Exams	
	Fever	Gastrointestinal	Hemodynamical	Hematological	Rheumatologic	Dermatological	Other	Inflammation Markers Elevated (Ferritin, CRP)	Elevated IL-18
c.620G>A, p.Arg207Lys^Case 1^	+	+	+	+	−	+	−	+	+
c.1010C>A, p.Thr337Asn^Case 2^	+	+	−	+	−	−	−	+	NA
c.1009A > T, p.Thr337Asn [5]	+	−	−	+	−	−	−	+	NA
c.1022T>C, p.Val341Ala [6,7]	+	+	−	+	+	+	Respiratory (acute respiratory distress syndrome, Intra alveolar hemorrhage) /Renal (Acute failure)	+	+
c.1021G>C, p.Val341Leu [8]	+	+	+	+	−	+	−	+	+
c.1009A > T, p.Thr337Ser [9]	+	+	−	+	−	−	−	+	NA
c.512C> T p.Ser171Phe [10]	+	+	−	+	−	−	−	NA	NA
c.1589A>C, p. His443Pro [11]	+	−	−	−	+	+	−	−	NA
c.1333T>C p.Ser445Pro [12]	−	−	−	−	+	+	−	−	NA
c.514G>A p.Gly172Ser [13]	+	−	−	−	−	+	−	NA	NA
c.529A>G, p. Thr177Asn [14]	+	+	−	-	−	+	Neurological (mental retardation, aseptic meningitis, sensorineural deafness, brain atrophy)	NA	NA
c.1965G>C, p. Trp655Cys [15]	+	+	−	+	−	+	−	+	+
c.1970A> T, p.Gln657Leu [16]	+	+	−	+	−	+	−	+	+

+: symptom present ; −: absent ; NA: Non Available; Gastrointestinal symptoms reported: Bloating, diarrhea, failure to thrive, cholescystitis, enterocolitis, transaminasemia, gallbladder hydrops, abcess anal margin, ascite, intestinal intussusception, pain, vomiting; Hemodynamical symptom reported: Vasoplegic shock; Hematological symptoms reported: MAS, hepatosplenomegaly, cytopenia, hypereosinophilia, disseminated intravascular coagulation; Rheumatologic symptoms reported: myalgia, arthralgia; Dermatological symptoms reported: Urtical rash, erythematous nodules or psoriasis.

## Data Availability

The data presented in this study are available on reasonable request from the corresponding author.

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
