# Peer review of "NLRC4 GOF Mutations, a Challenging Diagnosis from Neonatal Age to Adulthood"

_jcm, 2021, doi:10.3390/jcm10194369_

Round 1
Reviewer 1 Report
The paper represents an interesting addition to the field by increasing numbers of cases. The paper would be strongly improved by making the case reports more concise. In addition, the description of the previously described cases could be shortened and better presented as a table. A short report would impart the same information and be much more readable.
Author Response
We thank the reviewer for his/ her insightful comments.
We have considerably reduced the presentation of clinical cases and we have introduced a table ( new table 2) allowing us to review previous clinical cases. As a result, this manuscript has been deeply reformatted.
We hope these changes will satisfy the reviewer.
Reviewer 2 Report
Title: I suggest to modify in “ NLRC4 gof mutations: a challenging diagnosis from neonatal age to adulthood” as shock is not always a clinical features and thus the previous title might be misleading
Figure 2: please add here or in the text the maximum dosage of Anakinra administered to the patient.
Pg 5 line 165: please correct the reference “by Bracaglia and al. (2015 Pediatr Rheumatol (HGMD))”
Please clarify that IL18 might be also elevated in systemic juvenile arthrtitis (see https://pubmed.ncbi.nlm.nih.gov/33128796/ and https://pubmed.ncbi.nlm.nih.gov/33200207/). Thus, the role of IL-18 might be to differentiate autoinflammatory diseases/primary HLH from other disease, such as FPIES or other allergic or infectious ones.
Supplementary table : “Change Rhumatological“; change “on one episode” in “single episode”
Author Response
We thank the reviewer for her/his insightful comments.
We feel that we have adressed all the issues raised by the reviewer and hope this new version will be acceptable for publication in the journal.
1/ Title: I suggest to modify in “ NLRC4 gof mutations: a challenging diagnosis from neonatal age to adulthood” as shock is not always a clinical features and thus the previous title might be misleading
We have changed the title accordingly.
Figure 2: please add here or in the text the maximum dosage of Anakinra administered to the patient.
We have now added the maximum dosage of anakinra, canakinumab and steroids in the new figure 1.
Pg 5 line 165: please correct the reference “by Bracaglia and al. (2015 Pediatr Rheumatol (HGMD))”
This reference has been updated as requested.
Please clarify that IL18 might be also elevated in systemic juvenile arthrtitis (see https://pubmed.ncbi.nlm.nih.gov/33128796/ and https://pubmed.ncbi.nlm.nih.gov/33200207/). Thus, the role of IL-18 might be to differentiate autoinflammatory diseases/primary HLH from other disease, such as FPIES or other allergic or infectious ones.
We thank the reviewer to make this point. We have now cited these two papers and clarify the interest of IL-18 for FPIES / sepsis or capillary leak syndrome differential diagnosis.
Supplementary table : “Change Rhumatological“; change “on one episode” in “single episode”
This supplementary table has been removed in the last version.
Round 2
Reviewer 1 Report
Much improved. Nice job with the changes.